# Channel Model and Performance Analysis for MIMO Systems with Single Leaky Coaxial Cable in Tunnel Scenarios

**DOI:** 10.3390/s22155776

**Published:** 2022-08-02

**Authors:** Kai Zhang, Guoxin Zheng, Hua Wang, Changming Zhang, Xianbin Yu

**Affiliations:** 1Research Institute of Intelligent Network, Zhejiang Lab, Hangzhou 311121, China; hua.wang@zhejianglab.com (H.W.); zhangcm@zhejianglab.com (C.Z.); xyu@zhejianglab.com (X.Y.); 2Key Laboratory of Specialty Fiber Optics and Optical Access Networks, Joint International Research Laboratory of Specialty Fiber Optics and Advanced Communication, Shanghai Institute for Advanced Communication and Data Science, Shanghai University, Shanghai 200444, China; gxzheng@staff.shu.edu.cn; 3School of Electric and Information Engineering, Zhejiang University, Hangzhou 310027, China

**Keywords:** channel model, multiple–input multiple–output (MIMO), leaky coaxial cable (LCX), tunnel, capacity

## Abstract

Due to the limited space in the tunnel environment, multiple-input multiple-output (MIMO) systems with double-port fed leaky coaxial cables (LCXs) can not only reduce the number of LCXs, but also improve the channel capacity. Based on the geometry based on single bonce (GBSB) and electromagnetic field radiation theory of LCX, a MIMO channel model with double–port fed LCX in a tunnel scenario is proposed in this paper. The channel impulse response (CIR) is derived, and verified with measurement results in terms of channel capacity. The distribution of channel capacity of single double-port fed LCX under different LCX lengths in the tunnel scenarios has also been analyzed in this work, and the distribution of channel capacity for the LCX–MIMO system with long LCX is predicted. The results show that the single double-port fed LCX–MIMO system outperforms the dipole antenna MIMO system with respect to channel capacity in the considered tunnel scenarios.

## 1. Introduction

Leaky coaxial cable (LCX) has the advantage of low longitudinal power attenuation, so it is widely used in the long term evolution Metro (LTE–M) system of urban subways. In addition, fully-automated driving and other applications require high channel capacity of communication systems. The multiple–input multiple–output (MIMO) system composed of LCX fed at one port has been applied in the subway tunnel scenario. However, the tunnel scenario space is limited and it cannot support a large number of LCX. Therefore, a MIMO system composed of LCX fed at double ports is an area of research interest, as it can effectively reduce the number of LCX by half.

The research on MIMO systems with double ports fed LCX is mainly focused on measurements and analysis in recent works. For example, Dr. Yafei Hou and his team carried out a lot of measurement campaigns on the MIMO wireless channel with two ports fed into two LCXs in a linear cellular environment (e.g., subway tunnel, underground mine tunnel, corridor, etc.), and analyzed the relationship between channel parameters (e.g., LCX spacing [1,2,3], position of receiving antenna relative to LCX [4,5,6]) and channel characteristics (e.g., channel capacity [7] and condition number [8]). In [9,10], the authors research the LCX–MIMO communication systems by taking measurements in both tunnel [9] and indoor environments [10], and analyzing the relationship between the condition number, channel capacity and LCX spacing. It is found that the LCX spacing has limited impact on the condition number and channel capacity.

In order to further study the factors that may affect the performance of an LCX MIMO wireless channel, Dr. Yafei Hou and his team studied the impact of user location information on the performance of an LCX MIMO wireless channel [11,12,13,14]. A composite leaky cable based on user location information and a low complexity power allocation scheme is proposed for a 4 × 4 MIMO system in [11]. The results showed that the error of user location information could lead to loss of channel capacity for an LCX MIMO system [13,14].

Up to now, there are few theoretical studies on MIMO wireless channels with double-port fed LCX. In [15], Dr. Yiming Wu and his team proposed a channel model of a MIMO wireless communication system with two ports fed into a single LCX in open space. The theoretical expressions of channel impulse response (CIR) and channel correlation are derived. The correctness of the channel model is verified by Monte Carlo simulation, and the effects of channel correlation on different channel parameters settings and different LCX slots periods (*d*_0_ = 0.4~2.4*λ*)) have been analyzed. The results showed that the channel correlation changes periodically with the change of slot period.

According to the statistical properties of physical parameters in communication environment, the modelling of geometry–based on the single bounce (GBSB) model can be established by using basic theory in communication fields. Since the modelling of GBSB is simple and adaptable, it is therefore the most commonly used channel model theory in different scenarios [16,17,18,19,20].

To the best of the author’s knowledge, the MIMO technology of double-port fed LCX has not been applied to real scenarios yet, and the research on the MIMO wireless channel of double-port fed LCX in a tunnel scenario is mainly focused on the measurement. There is a lack of research on the channel model and its related performance analysis.

In this paper, we for the first time apply the GBSB theory and electromagnetic field radiation theory of the LCX to LCX–MIMO channel model and propose a novel 3D GBSB channel model. It is assumed that all the effective scatterers in the tunnel are distributed on the surface of the side wall and the roof. Based on this model, we investigate the impact of several parameters on channel capacity, e.g., receiver location in the tunnel, length of LCX, and element spacing of the receiving antenna array. Numerical results have shown that the performance of the LCX–MIMO system is better than that of the dipole antennas MIMO system in tunnel scenarios. The proposed model has been validated by comparing the theoretical channel capacity with measurement data.

The remainder of this paper is organized as follows. In Section 2, the MIMO system model with double ports fed LCX and the radiation characteristics of LCX are described. The expressions of CIR for a MIMO system with double ports fed LCX are derived in Section 3. The MIMO channel measurement and performance analysis of double-port fed LCX are carried out in Section 4. Finally, some conclusions are drawn in Section 5.

## 2. System Model of LCX MIMO and Radiation Characteristics of LCX

The MIMO system with double-port fed LCX in a tunnel is depicted in Figure 1b. The transmitter uses double-port fed LCX (the LCX structure for vertical slots is depicted in Figure 1a), which is installed on the side wall of the tunnel, where S1 and S2 denote the input signals of double ports for LCX, respectively. The two dipole antennas at the receiver (i.e., Rx1 and Rx2) are both located in the middle of the ground in the tunnel, and the element spacing of the receiving antennas array is denoted by *D* in Figure 1c. The polarization mode of Rx is horizontal polarization in Figure 1b. The coordinates of Rx1 are (*x*_0_, *y*_0_, *z*_0_), and the horizontal distance between the receiving antenna array and the LCX is *x*_0_. *p* denotes the period of LCX slots. *S_m_* denotes the *m*-th (*m* = 1, …, *M*, *M* is the total number of scatterers) scatterer in the side wall or the roof of the tunnel (i.e., Figure 1c,d). The origin of the coordinate system is the feed port S1 of the LCX. The *z*-axis is the axial direction of LCX, the cross section of the tunnel is *xoy* plane, in which the *x*-axis is the horizontal direction and the *y*-axis is the vertical direction. The other parameters of the LCX–MIMO system are shown in Table 1 and Table 2.

To avoid the interference of high-order harmonics, LCX with single-mode is considered in this paper. Figure 2 illustrates the electromagnetic wave signal radiation modes of the MIMO system with single-port fed LCX and double-port fed LCX, respectively.

According to the electromagnetic properties of LCX, the main radiation direction of LCX in the single-mode method *δ*_−1_ can be expressed as
(1)$$\delta_{-1}=\sin^{-1}\left(\sqrt{\varepsilon_{r}}-\frac{\lambda }{P}\right)$$
where, *λ* = *c/f* denotes the wavelength of the carrier wave LCX and *c* denotes the velocity of light.

Due to the roughness of the surface on the side wall and the roof in the tunnel, there are a large number of reflections and scatterings in the propagation of electromagnetic waves. Assuming that there are a large number of scatterers on the side wall and roof surface of the tunnel (as shown in Figure 1d), there are both line-of-sight (LoS) paths and non-LoS (NLoS) paths in the propagation of electromagnetic waves in the tunnel. According to the theory of electromagnetic field radiation characteristics of single-port fed LCX in the tunnel, as described in [21,22], at a certain point the radiation field of a slot of LCX in the tunnel includes both LoS and NLoS radiation field. The strength of electric field can be expressed as
(2)$$E_{z}^{LoS}=E_{0}\frac{e^{-jk_{0}r_{0}}}{r_{0}}\sin\left(\theta_{0}+\delta_{-1}\right)$$
(3)$$E_{z}^{NLoS}=E_{0}\frac{e^{-jk_{0}\left(r_{1}+r_{2}\right)}}{r_{1}r_{2}}\sin\left(\theta_{1}+\delta_{-1}\right)$$
where *E*_0_ denotes the electric strength of the LCX slot, *r*_0_ denotes the distance between a certain point in the radiation field of a slot of LCX and a slot of LCX, *r*_1_ is the distance from a slot of LCX to one of the scatterers at the side wall or the roof surface in the tunnel, *r*_2_ is the distance from one of scatterers at the side wall or the roof surfaces in the tunnel to a certain point in the radiation field of a slot of LCX, *θ*_0_ denotes the angular between *r*_0_ and axis direction of LCX, *θ*_1_ denotes the angular between *r*_1_ and axis direction of LCX, *k*_0_ = 2π*f*/*c* is the wave number of free-space, *f* is carrier frequency, and *c* is the velocity of light in free-space.

According to Table 2, the length of LCX is *L*, the period of LCX slot is *p*, and the total number of slots is *n* = floor (*L*/*p*)−1. By defining the longitudinal power attenuation constant of LCX as *α* (dB/100 m), the amplitude attenuation of the adjacent slot of LCX is $\alpha_{0}=10^{-\alpha P/2/10/100}$, and the amplitude attenuation and phase shift for the *i*-th (*i* = 1, …, *n*) slot of LCX can be expressed as
(4)$$\alpha_{i}=\alpha_{0}^{i}$$
(5)$$\beta_{i}=k_{r}iP$$
(6)$$k_{r}=k_{0}\sqrt{\varepsilon_{r}}$$
where, *k_r_* denotes the wave number in LCX.

Since a single LCX has *n* slots, according to Formulas (1)–(6), the electric field strength of the LoS path and NLoS path of LCX in the tunnel can be expressed as
(7)$$E_{z}^{LoS}=\sum_{i=1}^{N}E_{i}\frac{e^{-jk_{0}r_{i}}}{r_{i}}\sin\left(\theta_{i}+\delta_{-1}\right)$$
(8)$$E_{z}^{NLoS}=\sum_{i=1}^{N}\sum_{m=1}^{M}E_{i}\frac{e^{-jk_{0}\left(r_{im,1}+r_{m,2}\right)}}{r_{im,1}r_{im,2}}\sin\left(\theta_{im}+\delta_{-1}\right)$$
where, $E_{i}=E_{0}\alpha_{i}e^{-j\beta_{i}}$ is the electric field strength of the *i*-th slot of LCX, *E*_0_ denotes the input electric filed strength of LCX, *r_i_* denotes the distance from the *i*-th slot of LCX to a certain point in the slot electric field, *r_im_*_,1_ denotes the distance from the *i*-th slot of LCX to the *m*-th (*m* = 1, …, *M*) scatterer on the side wall or the roof of the tunnel, *r_im_*_,2_ denotes the distance from the *m*-th scatterer on the side wall or the roof of the tunnel to a certain point in the slot electric field, *θ_i_* denotes the angle between *r_i_* and the axis direction of LCX, *θ_im_* denotes the angle between *r_im_*_,1_ and axis direction of LCX.

## 3. CIR for the LCX-MIMO System

### 3.1. CIR of LoS Path Only

The MIMO system with single double-port fed LCX considering only an LoS path is described in Figure 3, and most settings of the LCX-MIMO system are the same as those in Figure 1c.

According to Formulas (4)–(7), the CIR expressions of the LoS path from the input signals S1 and S2 of double ports feed LCX at transmitter (Tx) to the first receiving antenna Rx1 and can be expressed as
(9)$$h_{S1,1}=\sum_{i=1}^{N}\alpha_{i}e^{-j\beta_{i}}\cdot \frac{\sin\left(\theta_{1,i}+\delta_{-1}\right)}{r_{1,i}}\cdot e^{-jk_{0}r_{1,i}}$$
(10)$$h_{S2,1}=\sum_{i=1}^{N}\alpha_{\left(N-i+1\right)}e^{-j\beta_{\left(N-i+1\right)}}\cdot \frac{\sin\left(\theta_{1,i}-\delta_{-1}\right)}{r_{1,i}}\cdot e^{-jk_{0}r_{1,i}}$$
where *r*_1,*i*_ denotes the distance of the LoS path from the *i*-th slot of LCX to Rx1, *θ*_1,*i*_ denotes the angle between *r*_1,*i*_ and axis direction of LCX.

### 3.2. CIR of Both LoS Path and NLoS Path

The MIMO system with single double-port fed LCX considering both an LoS path and NLoS path is depicted in Figure 4. Most settings of the LCX-MIMO system are the same as those in Figure 1c, and the coordinates of the *m*-th scatterer on the side wall or the roof of the tunnel are (*x_m_*, *y_m_*, *z_m_*).

According to Formulas (4)–(8), the CIR expressions of the LoS path and NLoS path from the input signals S1 and S2 of double-port fed LCX at transmitter (Tx) to first receiving antenna Rx1 can be expressed as
(11)$$h_{S1,1}^{LoS}=\underset{N\leftarrow \infty }{\lim}\sqrt{\frac{\Omega_{S1,1}K_{S1,1}}{N\left(K_{S1,1}+1\right)}}\sum_{i=1}^{N}g_{S1,1,i}^{LoS}\cdot e^{-j\beta_{i}}\cdot e^{-jk_{0}r_{1,i}}$$
(12)$$h_{S1,1}^{NLoS}=\underset{\begin{array}{l} N\to \infty \\ M\to \infty \end{array}}{\lim}\sqrt{\frac{\Omega_{S1,1}}{NM\left(K_{S1,1}+1\right)}}\sum_{i=1}^{N}\sum_{m=1}^{M}\left(\begin{array}{l} g_{S1,1,im}^{NLoS}\cdot e^{-j\beta_{i}}\cdot \\ e^{-j\left[k_{0}\left(r_{1,im}^{\prime }+r_{1,im}^{\prime \prime }\right)-\varphi_{im}\right]} \end{array}\right)$$
(13)$$h_{S2,1}^{LoS}=\underset{N\leftarrow \infty }{\lim}\sqrt{\frac{\Omega_{S2,1}K_{S2,1}}{N\left(K_{S2,1}+1\right)}}\sum_{i=1}^{N}g_{S2,1,i}^{LoS}\cdot e^{-j\beta_{\left(N-i+1\right)}}\cdot e^{-jk_{0}r_{1,i}}$$
(14)$$h_{S2,1}^{NLoS}=\underset{\begin{array}{l} N\to \infty \\ M\to \infty \end{array}}{\lim}\sqrt{\frac{\Omega_{S2,1}}{NM\left(K_{S2,1}+1\right)}}\sum_{i=1}^{N}\sum_{m=1}^{M}\left(\begin{array}{l} g_{S2,1,im}^{NLoS}\cdot e^{-j\beta_{\left(N-i+1\right)}}\cdot \\ e^{-j\left[k_{0}\left(r_{1,im}^{\prime }+r_{1,im}^{\prime \prime }\right)-\varphi_{im}\right]} \end{array}\right)$$
where Ω_S1,1_ = E [|*h*_S1,1_|^2^] and Ω_S2,1_ = E [|*h*_S2,1_|^2^] denote the power of CIR from the input signal S1 to Rx1 and the power of CIR from input signal S2 to Rx2, respectively. The expressions of *h*_S1,1_ and *h*_S2,1_ are $h_{S1,1}=h_{S1,1}^{LoS}+h_{S1,1}^{NLoS}$ and $h_{S2,1}=h_{S2,1}^{LoS}+h_{S2,1}^{NLoS}$, respectively. *K_j_*_,1_ (*j* = S1, S2) denotes Rice *K*-factor, which is defined as the ratio between the CIR power of the LoS path and the CIR power of the NLoS path, i.e., $K_{j,1}={\left|h_{j,1}^{LoS}\right|}^{2}/E\left[{\left|h_{j,1}^{NLoS}\right|}^{2}\right]$. *r*_1,*i*_ denotes the LoS path distance from the *i*-th slot of LCX to Rx1. $r_{1,im}^{\prime }$ denotes the distance from the *i*-th slot of LCX to *S_m_*. $r_{1,im}^{"}$ denotes the distance from *S_m_* to Rx1. *φ_im_* denotes the random phase with uniform distribution in the range of [0, 2*π*) by the *m*-th scatterer. $g_{S1,1,i}^{LoS}$, $g_{S1,1,i}^{NLoS}$, $g_{S2,1,i}^{LoS}$, and $g_{S2,1,i}^{NLoS}$ denote the channel gain of LoS path and NLoS path from input signals S1 and S2, respectively, and they can be calculated as
(15)$$g_{S1,1,i}^{LoS}=\frac{\frac{\alpha_{i}}{r_{1,i}}\sin\left(\theta_{1,i}+\delta_{-1}\right)}{\sqrt{N^{-1}\sum_{i=1}^{N}{\left\{\frac{\alpha_{i}}{r_{1,i}}\sin\left(\theta_{1,i}+\delta_{-1}\right)\right\}}^{2}}}$$
(16)$$g_{S1,1,i}^{NLoS}=\frac{\frac{\alpha_{i}}{r_{1,im}^{\prime }r_{1,im}^{\prime \prime }}\sin\left(\theta_{1,im}+\delta_{-1}\right)}{\sqrt{{\left(NM\right)}^{-1}\sum_{i=1}^{N}\sum_{m=1}^{M}{\left\{\frac{\alpha_{i}}{r_{1,im}^{\prime }r_{1,im}^{\prime \prime }}\sin\left(\theta_{1,im}+\delta_{-1}\right)\right\}}^{2}}}$$
(17)$$g_{S2,1,i}^{LoS}=\frac{\frac{\alpha_{\left(N-i+1\right)}}{r_{1,i}}\sin\left(\theta_{1,i}-\delta_{-1}\right)}{\sqrt{N^{-1}\sum_{i=1}^{N}{\left\{\frac{\alpha_{\left(N-i+1\right)}}{r_{1,i}}\sin\left(\theta_{1,i}-\delta_{-1}\right)\right\}}^{2}}}$$
(18)$$g_{S2,1,i}^{NLoS}=\frac{\frac{\alpha_{\left(N-i+1\right)}}{r_{1,im}^{\prime }r_{1,im}^{\prime \prime }}\sin\left(\theta_{1,im}-\delta_{-1}\right)}{\sqrt{{\left(NM\right)}^{-1}\sum_{i=1}^{N}\sum_{m=1}^{M}{\left\{\frac{\alpha_{\left(N-i+1\right)}}{r_{1,im}^{\prime }r_{1,im}^{\prime \prime }}\sin\left(\theta_{1,im}-\delta_{-1}\right)\right\}}^{2}}}$$
where *θ*_1,*i*_ denotes the angle between *r*_1*,i*_ and axis direction of LCX, and *θ*_1,*im*_ denotes the angle between $r_{1,im}^{\prime }$ and axis direction of LCX.

## 4. Measurement and Performance Analysis of the LCX-MIMO Channel

### 4.1. Measurement Scenario for MIMO Channel with Double-Port Fed of Single LCX

The measurement scenario is described in Figure 5. The measurement is performed in a tunnel provided by Zhongtian Technology Company (ZTT) in Nantong. The total length of the tunnel is 100 m, of which the length of the rectangular tunnel and the arch tunnel are both 50 m. The LCX used in measurement and is provided by ZTT. In this paper, we only perform measurement in the rectangular tunnel. The LCX used in measurement is a vertical periodic slot, and the polarization mode of LCX is horizontal polarization. As such, the polarization mode of all receiving antennas is horizontal polarization. Due to physical limitations, the virtual antenna array is adopted at Rx. It can be seen in Figure 5d that the element spacing of the virtual antenna array at Rx is 0.5 m. The LCX in the measurement system is installed on one side of the wall in the tunnel, and the receiving antenna is installed on a trolley with guided rail in the tunnel. The other parameters related to the measurement scenario and LCX are listed in Table 3 and Table 4.

### 4.2. Rice K-Factor

In order to make the parameter setting of the theoretical analysis consistent with the measurement, the Rice factor used in the theoretical analysis is obtained from the measurement data. The Rice *K*-factor (i.e., *K*-factor) has great significance in channel characterization of various wireless communication scenarios, which can be calculated as [23]
(19)$$K=\frac{E{\left[\left|H\right|\right]}^{2}}{2\mathrm{var}\left(\left|H\right|\right)}$$
where E [•] and var (•) denote the mean and the variance of random variable, respectively.

The cumulative distribution function (CDF) of the estimated *K*-factor from the measurement data is shown in Figure 6. Table 5 lists the maximum, the minimum and the mean value of the estimated *K*-factor. The mean value of *K*-factor, which is 4.3819 dB from the measurement, will be used in the theoretical analysis.

### 4.3. Channel Capacity for MIMO System with Single Double-Port Fed LCX

#### 4.3.1. Channel Capacity

The channel matrix of a MIMO system with a single double-port fed LCX can be expressed as
(20)$$H=\left[\begin{array}{cc} h_{S1,1} & h_{S1,2}\\ h_{S2,1} & h_{S2,2} \end{array}\right]$$
where $h_{lp}=h_{lp}^{LoS}+h_{lp}^{NLoS}$ (*p* = 1,2; *l* = S1, S2).

In this paper, the equal power (EP) allocation method is adopted for power allocation. According to Shannon’s theorem, the expression of MIMO channel capacity can be calculated as
(21)$$C_{EP}=\log_{2}(I+\frac{SNR}{n_{T}}H^{H}\cdot H)\;[\mathrm{bit}/s/\mathrm{Hz}]$$
where **I** is a *n**_Rx_* × *n**_Rx_* (*n**_Rx_* = 2) unity matrix. **H** is a *n**_Tx_* × *n**_Rx_* (*n**_Tx_* = *n**_Rx_* = 2) channel matrix. SNR is the mean signal to noise ratio at Rx. (•)*^H^* denotes the conjugate transpose operation.

#### 4.3.2. Performance Analysis

In this paper, multipath extraction of measurement data is carried out by a sliding correlation algorithm. The SNR and *K*-factor are set to 10 dB and 4.3 dB, respectively, for channel capacity analysis. To verify the correctness of the MIMO channel model with double-port fed LCX, the channel capacity from theoretical calculation and measurements are plotted in Figure 7 and Table 6. It is found during the study that, when the number of effective scatterers *M* in the tunnel is less than 200, the channel capacity changes significantly with *M*. However, when *M* is greater than 200, the channel capacity does not change significantly with *M*. Therefore, the number of effective scatterers *M* in tunnel is set to be 200 in the theoretical analysis.

It can be seen from Figure 7 and Table 6 that the theoretical analysis of the channel capacity for a MIMO wireless system with single double-port fed LCX with LoS only path and both LoS and NLoS paths match quite well with the measurements. The small difference may be caused by human factors in the measurement process and the change of polarization characteristics in the process of signal propagation in the tunnel. The MIMO channel capacity with LoS only path is slightly smaller as compared to the case with both LoS and NLoS paths, which is due to the multipath effect caused by the electromagnetic wave signal radiated from a series of periodically distributed slots of the leaky cable, and the reflection of electromagnetic waves on the inner wall of the tunnel increases the number of multipaths. The measurement results of the MIMO channel capacity of LCX are consistent with the theoretical results, and the distribution range is relatively close, which verifies the correctness of the derived MIMO wireless channel model with single double-port fed LCX.

#### 4.3.3. Prediction of MIMO Channel Capacity

It should be noted that most parameters of the LCX-MIMO system in this paper are set to be the same. In order to understand the distribution of MIMO channel capacity with double-port fed LCX in case the length of LCX increases, this section will predict the channel capacity of the MIMO system with long LCX, as shown in Figure 8. Most of the settings of the LCX-MIMO system are the same as those in Figure 1c. The range of z-axis for Area 1 is 0~50 m, the range of z-axis for Area 2 is 225~275 m. Other parameters of LCX MIMO system are listed in Table 7.

According to Formulas (9)–(18), (20) and (21), the theoretical results of the channel capacity for a MIMO system with double-port fed LCX considering LoS only path and both LoS path and NLoS path are shown in Figure 9, Figure 10 and Figure 11.

In Figure 9, it is shown that: (1) the capacity of the LCX MIMO system considering LoS only propagation path and both LoS and NLoS propagation paths at area 1 of the tunnel is larger than that at area 2 of the tunnel; (2) there is little difference for capacity of the LCX MIMO system between considering LoS only propagation path and both LoS and NLoS propagation paths at area 1 and area 2 of the tunnel.

According to the results of Figure 7 and Figure 9, it is found that there is little difference for the capacity of the LCX MIMO system between considering LoS only propagation path and both LoS and NLoS propagation paths at different areas in the tunnel. Therefore, by only considering the LoS propagation path for the LCX MIMO system in Figure 10, it is found that: at different areas in the tunnel, the capacity of the LCX-MIMO system at double ports areas of LCX is higher than that at the middle of LCX. The reason is that the radiation power of the slot at the double ports of the LCX is greater than at the middle of the LCX, when the signals are fed at the double ports of the LCX. Due to the great difference in power between the two signals transmitted from the far distance and the near distance in the signal feeding area for the double-port fed LCX, the RF (radio frequency) front port requires a large linear working range and proprietary technology should be adopted to decompose the two signals.

Figure 11 presents the channel capacity of the LCX-MIMO system with single double-port fed LCX under different areas in the tunnel and different element spacing at receiving antenna array. It is found that: (1) the capacity of LCX-MIMO system changes periodically with the element spacing of receiving antenna array increasing at different areas in the tunnel; (3) the capacity of the LCX-MIMO system at area 1 in the tunnel is greater than at area 2 in the tunnel.

### 4.4. Analysis of Channel Capacity for MIMO System with Single-Port Fed LCX and Double-Port Fed LCX

We further explore the channel characteristics of the LCX-MIMO system with double-port fed LCX. Figure 12 shows the capacity of the LCX-MIMO system with single double-port fed LCX and two single-port fed LCXs. The LCX spacing of the LCX-MIMO system with two single-port fed LCXs is 3λ. The other parameters of LCX-MIMO system with single double-port fed LCX are the same as the LCX-MIMO system with two single-port fed LCXs. According to Figure 12, when the LCX-MIMO system consider LoS only propagation path and consider both LoS and NLoS propagation paths, the capacity of the LCX-MIMO system with single double-port fed LCX can reach or approach that of the LCX-MIMO system with two single-port fed LCXs for the same parameter configuration. It shows the feasibility of the LCX-MIMO system with single double-port fed LCX.

### 4.5. Channel Capacity Comparison for MIMO System with Different Transmitting Antennas

Here, we discuss the difference in the communication system performance between the LCX-MIMO system with single LCX fed into double ports and a dipole antennas MIMO system in a tunnel scenario. The LCX-MIMO system with single double-port fed LCX and the dipole antennas MIMO system are constructed in Figure 13. *S_m_* (*m* = 1, …, *M*, where *M* = 100 denotes the total number of scatterers at the side wall and roof in the tunnel) denotes *m*-th scatterer at the side wall and roof in the tunnel. Both the length of LCX in the LCX-MIMO system and the element spacing of the transmitting antenna array in the dipole antennas MIMO system are *l*0. For both the LCX-MIMO system and the dipole antennas MIMO system, the element spacing of the receiving antenna array is *D* = 0.5 m, the vertical distance and initial horizontal distance between receiver and transmitter are *y*_0_ = 1.1 m and 1.5 m, respectively. The *x*-axis and *z*-axis value range of Rx1 in the receiving antenna array are 1.5~2.5 m and 0~2 m, respectively. Most of the parameters of LCX are the same as Table 7.

According to Formulas (9)–(18), (20) and (21), the theoretical capacity of the LCX-MIMO system and the dipole antennas MIMO system for considering LoS only propagation path and considering both LoS and NLoS propagation paths are calculated as shown in Figure 14. By comparison, as shown in Figure 14a,b, there are several conclusions that can be drawn, which are as follows:When the system is considering only LoS propagation path and considering both LoS and NLoS propagation paths, the capacity of the dipole antennas MIMO system gradually increases with an increase in the element spacing of the transmitting antenna array, but the capacity of the LCX-MIMO system shows just a little change, with an increase in the length of LCX.For the dipole antennas MIMO system, the capacity with considering only the LoS propagation path is smaller than when considering both LoS and NLoS propagation paths, obviously.For the LCX-MIMO system, the difference in capacity between considering only the LoS propagation path and considering both LoS and NLoS propagation paths is little. It is shown that the performance of the LCX-MIMO system is better than that of the dipole antennas MIMO system.

## 5. Conclusions

In this paper, we proposed a channel model for the LCX-MIMO system with single double-port fed LCX. By comparing the measurement results with the theoretical results of the capacity when considering the LoS only path and both LoS and NLoS paths, the validation and feasibility of the proposed channel model has been verified. The capacity of the LCX-MIMO system with single double-port fed LCX for a different length of LCX in the tunnel scenario has been analyzed. The capacity of the LCX-MIMO system with single double-port fed LCX and two single-port fed LCX under the same configuration has been analyzed and compared, as well as the capacity of the LCX-MIMO system and dipole antennas MIMO system. Based on the numerical results, it is found that:When the tunnel and LCX is very long, the capacity of the LCX-MIMO system at the double ports area of LCX in the tunnel is higher than that at the middle area of the LCX.For the LCX-MIMO system with a single double-port fed LCX, the capacity of system fluctuates periodically when changing the element spacing of the receiving antenna array.With the same configurations, the capacity of the LCX-MIMO system with single double-port fed LCX can reach or approach that of the LCX-MIMO system with two single-port fed LCX.In the considered tunnel scenario, the system performance of LCX-MIMO system with single double-port fed LCX is much better than that of th dipole antennas MIMO system.

## Figures and Tables

**Figure 1 sensors-22-05776-f001:**
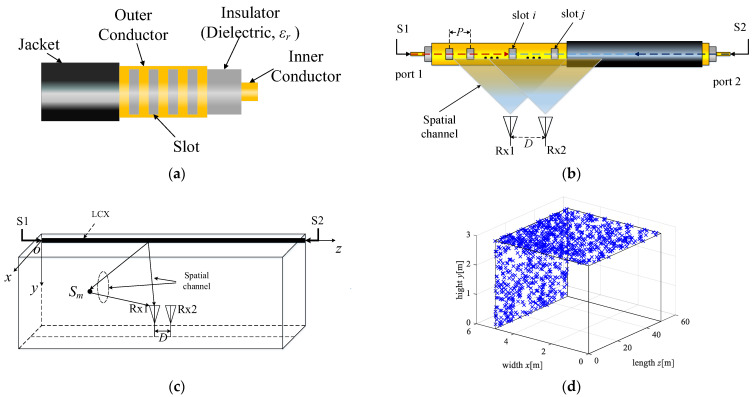
(**a**) LCX structure for vertical slots; (**b**) LCX-MIMO system; (**c**) propagation path of signal for LCX–MIMO system; (**d**) the distribution of effective scatterer in tunnel, “×” denotes the scatterer in the side wall or the roof of the tunnel.

**Figure 2 sensors-22-05776-f002:**
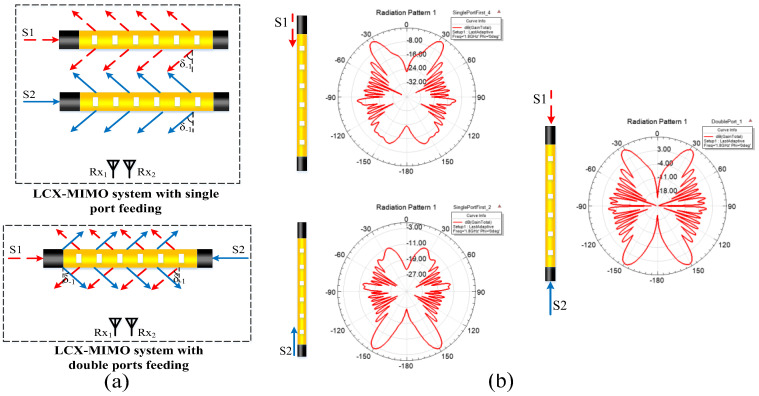
(**a**) Radiation modes of the electromagnetic wave signal for the LCX−MIMO system; (**b**) simulation radiation patterns of LCX by HFSS software (center frequency is 1.8 GHz, period of LCX slot is 0.25 m).

**Figure 3 sensors-22-05776-f003:**
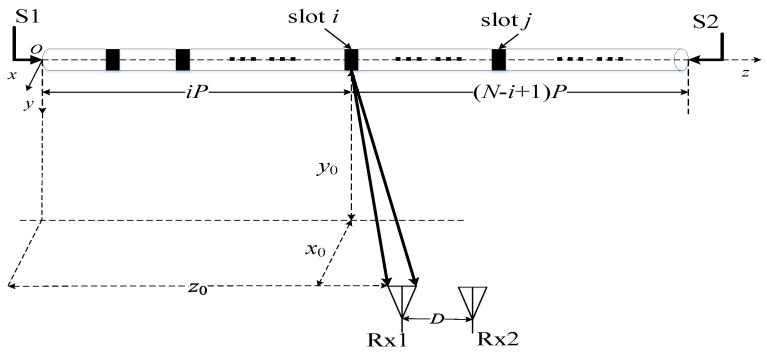
MIMO system with double-port fed of single LCX considering only an LoS path.

**Figure 4 sensors-22-05776-f004:**
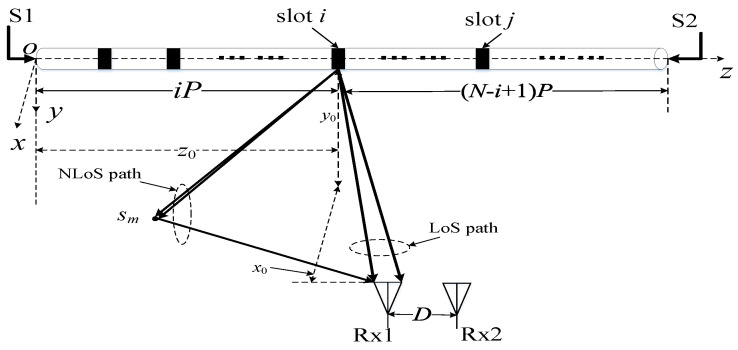
MIMO system with single double-port fed LCX considering both LoS and NLoS paths.

**Figure 5 sensors-22-05776-f005:**
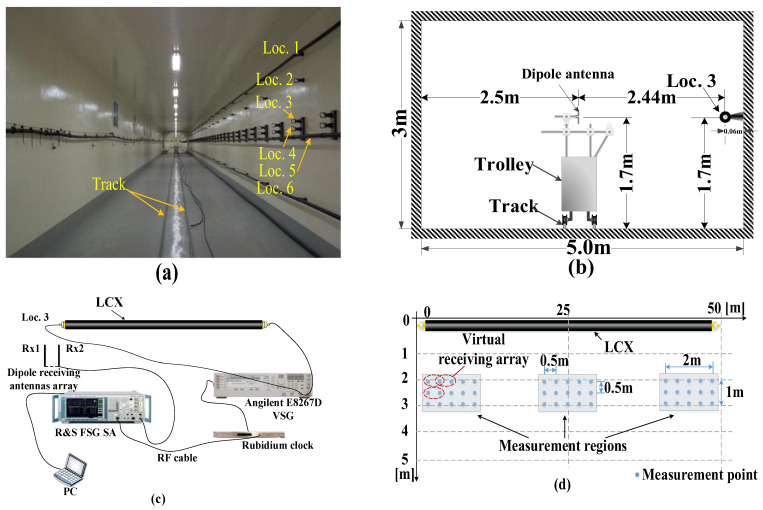
Description of the measurement: (**a**) the tunnel scenario used for measurement; (**b**) cross section view of the tunnel; (**c**) connection diagram of LCX MIMO system; (**d**) receiving antenna location.

**Figure 6 sensors-22-05776-f006:**
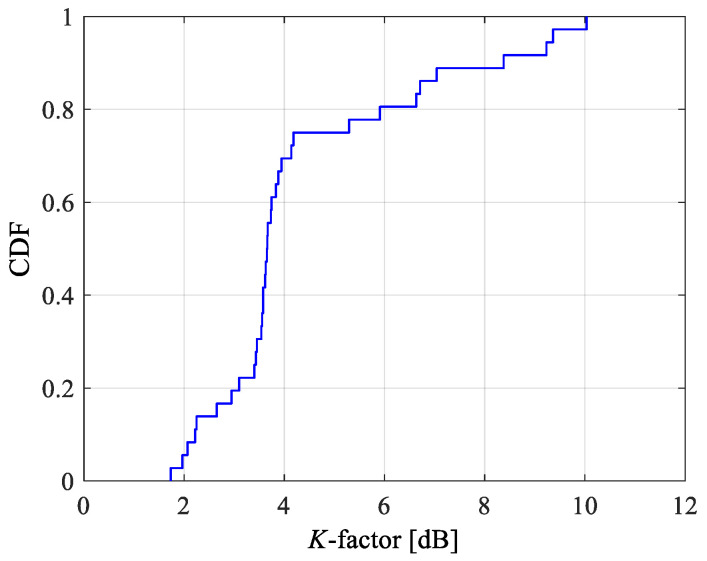
Cumulative distribution function (CDF) of *K*-factor from LCX MIMO channel measurements.

**Figure 7 sensors-22-05776-f007:**
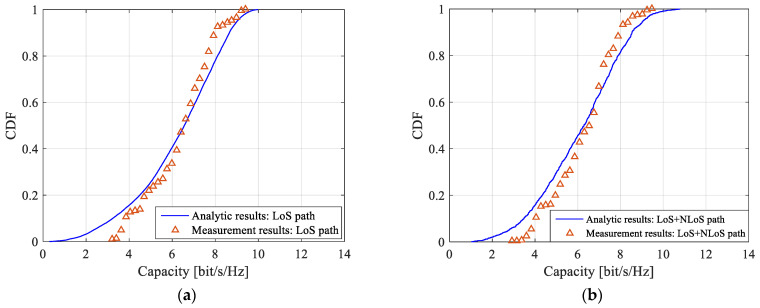
CDF of channel capacity: (**a**) LoS only path; (**b**) both LoS and NLoS paths.

**Figure 8 sensors-22-05776-f008:**
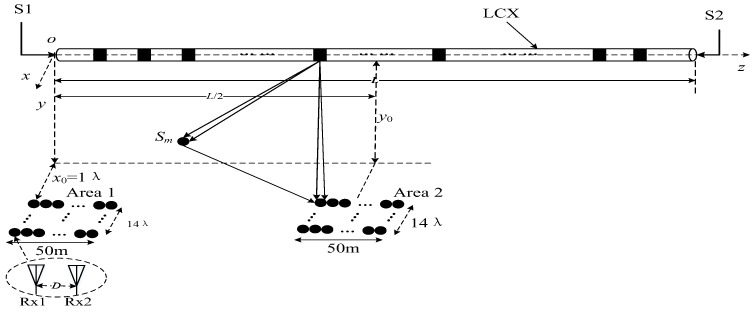
MIMO system with single long double-port fed LCX in tunnel.

**Figure 9 sensors-22-05776-f009:**
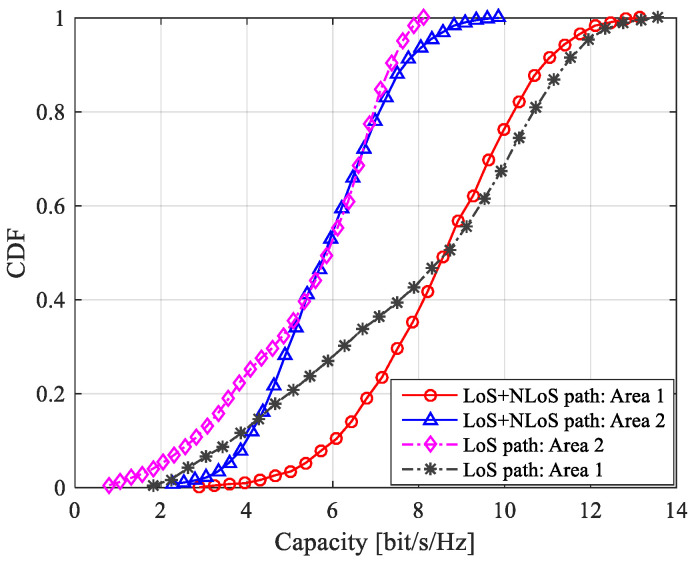
CDF of channel capacity for a MIMO system with double-port fed LCX considering LoS only path and both LoS and NLoS paths for different locations in the tunnel.

**Figure 10 sensors-22-05776-f010:**
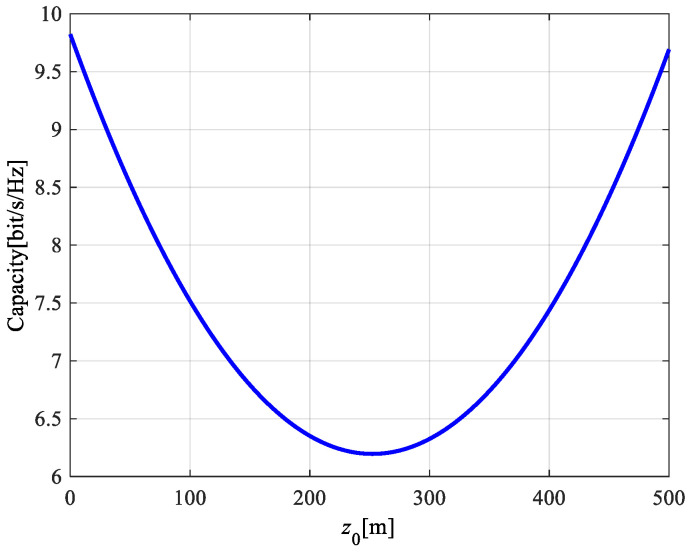
Channel capacity of the LCX-MIMO system with a different location in the tunnel.

**Figure 11 sensors-22-05776-f011:**
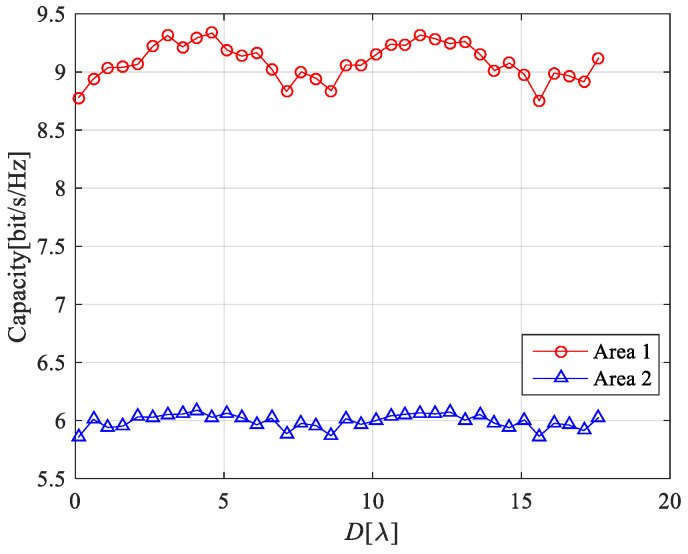
Channel capacity of the LCX-MIMO system for LoS only path.

**Figure 12 sensors-22-05776-f012:**
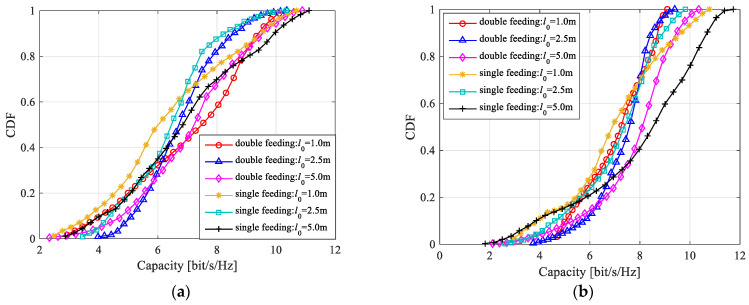
Channel capacity of the MIMO system for double-port fed and single-port fed LCX: (**a**) LoS propagation path; (**b**) both LoS and NLoS propagation paths.

**Figure 13 sensors-22-05776-f013:**
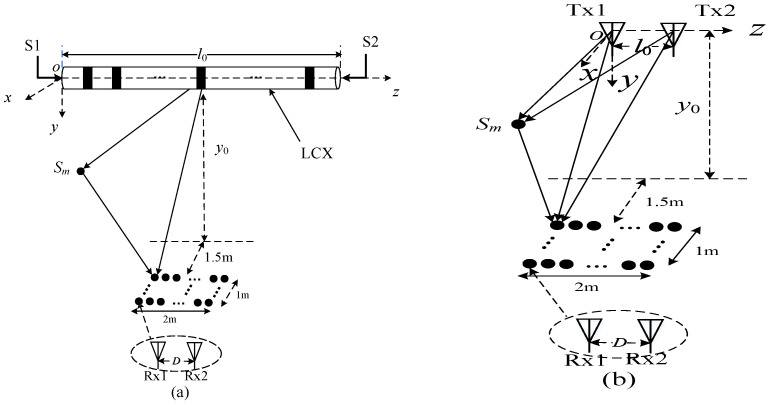
MIMO system: (**a**) transmitting antenna and receiving antenna are LCX and dipole antennas, respectively; (**b**) transmitting antenna and receiving antenna are both dipole antennas.

**Figure 14 sensors-22-05776-f014:**
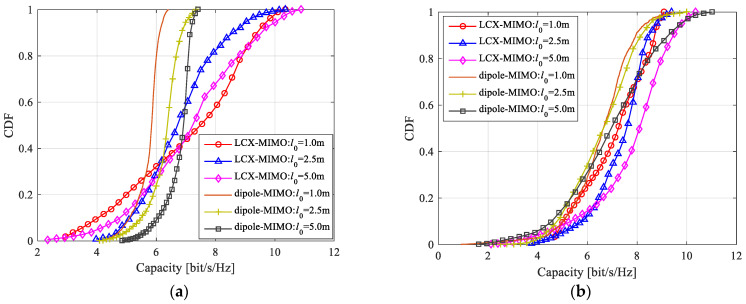
Channel capacity for LCX MIMO system and dipole antennas MIMO system: (**a**) LoS only path; (**b**) both LoS and NLoS paths.

**Table 1 sensors-22-05776-t001:** Parameters for LCX–MIMO system.

Description	Symbol
Carrier frequency	*f*
Coordinate of *m*-th scatterer	(*x_m_*, *y_m_*, *z_m_*)
Height of LCX	*h*
element spacing of receiving antennas array	*D*
Coordinate of Rx1	(*x*_0_, *y*_0_, *z*_0_)
Horizontal distance between LCX and receiving antennas array	*x* _0_
Size of tunnel (length × width × height)	*L* × *W* × *H*
Total number of scatterers	*M*

**Table 2 sensors-22-05776-t002:** Parameters for LCX.

Description	Symbol
Length of LCX	*L*
Longitudinal power attenuation within LCX	*α*
Period of LCX slot	*p*
Relative permittivity of LCX	*ε_r_*
Total number of LCX slot	*n*

**Table 3 sensors-22-05776-t003:** Parameters setup of measurement scenario.

Name	Value
Size of tunnel (*L* × *W* × *H*)	50 × 5 × 3 (m^3^)
Vertical height from LCX to tunnel ground (*h*)	1.7 m
Vertical height from receiving antenna to tunnel ground(*y*_0_)	1.7 m
Horizontal distance from LCX to receiving antenna (*x*_0_)	2.44 m
Carrier frequency (*f*)	1.8 G Hz
Wave length (*λ*)	0.167 m
Bandwidth (*B*)	40.8 MHz
Sample rate (*f_s_*)	81.6 MHz
Power of transmitting (*P*_Tx_)	20 dBm
Gain of receiving antenna *(G*_Rx_)	2.15 dBi

**Table 4 sensors-22-05776-t004:** Parameters of LCX.

Name	Value
Length of LCX (*L*)	50 m
Period of slot (*p*)	0.25 m
Width of slot (*w*)	8 mm
Materials of dielectric	polythene
Relative permittivity of dielectric (*ε_r_*)	1.25
Shape of slot	rectangular slot
Longitudinal power attenuation constant (*α*)	4.0 dB/100 m
Thickness of outer conductor (*d*_outer_)	0.35 mm
Thickness of inner conductor (*d*_inner_)	0.8 mm

**Table 5 sensors-22-05776-t005:** Estimate values of *K*-factor from the measurements.

Value	Maximum Value	Minimum Value	Mean Value
*K*-factor	10.0369 dB	1.7330 dB	4.3819 dB

**Table 6 sensors-22-05776-t006:** Mean value of channel capacity.

Propagation Path	Measurement	Theoretical
LoS path	6.3540 bit/s/Hz	6.0682 bit/s/Hz
Both LoS and NLoS paths	6.4314 bit/s/Hz	6.2478 bit/s/Hz

**Table 7 sensors-22-05776-t007:** Parameters of the LCX MIMO System.

Name	Value
Length of the tunnel and the LCX (*L*)	500 m
Element spacing of receiving antennas array (*D*)	0.5 m
Vertical distance from LCX to receiving antennas array (*y*_0_)	1.0 m
Number of scatterers (*M*)	500

## Data Availability

Not applicable.

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
