# Peer review of "Channel Model and Performance Analysis for MIMO Systems with Single Leaky Coaxial Cable in Tunnel Scenarios"

_sensors, 2022, doi:10.3390/s22155776_

Round 1

Reviewer 1 Report

Comments for Transmittal to Author:

1. Mention the methodology to incorporate the slot for leaky coaxial cables (LCXs) MIMO system.

2. Please improve the clearly of Fig. 2. and Fig. 5.

3. Include some latest references and compare with your works.

4. Check the eqn. (20), elements of matrix.

5. Provide the gain of the design LCXs MIMO antenna.

6. Many references are from same group. Why? 

Author Response

Please see the attachment. Thank you  very much for your comments and suggestions.

Reviewer 2 Report

The paper presents the channel model and performance analysis for MIMO systems with a Single LCX cable in the tunnels. The authors have a long history of publishing their work on are of LCX cables and MIMO systems. 

Sections 1 - 3  and partially 4 recap their published previously in different journals and conferences.  The extension of previous work to double feeding is straightforward, thus the contribution of this paper is marginal. There might be a problem with the copyright of some figures appearing in the author's previous publications.

However, there is some novelty in the paper, so it may be considered for publication.

But there are several mistakes in the paper, which I would not expect from so renominated authors. I will not mention all of them, because there are too many. For example usage of different symbols for mean/average (epsilon, E), missing indices of symbols within the text, not consistent referencing of the equations, tables, etc. tables are too large, and many other.

I suggest the authors read the text carefully because now it looks to be copied from different sources and sent to MDPI. And resubmit the paper.

Author Response

(The authors gave the same response as above.)

Round 2

Reviewer 1 Report

Please check the eqn. (20), elements of matrix is not correct. It is my suggestion:

[hs11   hs12

hs21    hs22]

Author Response

Thank you very much for your comment and suggestion. I modified the eqn. (20). Please see the attachment.

Reviewer 2 Report

I do not have any further comments. Accept paper in current form.

Author Response

Thank you very much for your warm work earnestly. Please see the attachment.
